# Key Factors Affecting the Flesh Flavor Quality and the Nutritional Value of Grass Carp in Four Culture Modes

**DOI:** 10.3390/foods10092075

**Published:** 2021-09-02

**Authors:** Junming Zhang, Gen Kaneko, Jinhui Sun, Guangjun Wang, Jun Xie, Jingjing Tian, Zhifei Li, Wangbao Gong, Kai Zhang, Yun Xia, Ermeng Yu

**Affiliations:** 1Guangdong Provincial Key Laboratory of Aquatic Animal Immune Technology, Pearl River Fisheries Research Institute, CAFS, Guangzhou 510380, China; 1904028161@stu.tjau.edu.cn (J.Z.); gjwang@prfri.ac.cn (G.W.); xj@prfri.ac.cn (J.X.); tianjj@prfri.ac.cn (J.T.); lzf@prfri.ac.cn (Z.L.); gwb@prfri.ac.cn (W.G.); zk@prfri.ac.cn (K.Z.); xy@prfri.ac.cn (Y.X.); 2Tianjin Key Lab of Aqua-Ecology and Aquaculture, Tianjin Agricultural University, Tianjin 300384, China; jhsun1008@tju.edu.cn; 3School of Arts & Sciences, University of Houston-Victoria, Victoria, TX 77901, USA; KanekoG@uhv.edu

**Keywords:** flavor quality, nutritional value, water quality factors, grass carp modes

## Abstract

Flavor and nutritional value are important qualities of freshwater fish products, but the key factors affecting these quality parameters remain unclear. In this study, four typical aquaculture modes, including the commercial feed treatment (control), faba bean treatment (FBT), grass powder treatment (GPT), and waving water treatment with commercial feed (WWT), were used to explore the regulatory effect of water quality and feed (eaten and uneaten) on the flesh flavor and nutrition in grass carp (*Ctenopharyngodon idella*), a freshwater fish of the largest global production. During the culture period (90 days), water quality parameters of the four modes were measured every 15 days, and the flavor quality was evaluated by volatile flavor compounds detection and electronic nose analyzer. Flesh crude protein, crude fat, free fatty acid and free amino acid profiles were also determined. The results showed that, in the late period, the FBT mode had the poorest water quality with highest concentrations of nitrite and nitrate, while the GPT mode has the best water quality among the four modes. Most flesh flavor compounds found in the flesh of the control, GPT and WWT modes were pleasant. In the FBT mode with the poorest water quality, on the other hand, we found lower flavor quality (higher contribution of fishy compounds), higher water content, and lower contents of crude protein, crude fat, free fatty acids and free amino acids, compared to the other three modes. Correlation analyses showed that nitrite and nitrate are probably key water quality factors affecting the flavor quality and nutritional values besides eaten feed and uneaten feed factors. This study can serve as an important reference for ecological regulation and feeding administration of flesh quality in freshwater aquaculture fish.

## 1. Introduction

Grass carp (*Ctenopharyngodon idella*), with the highest global production in freshwater species, has provided inexpensive and high-quality animal protein for consumers [1]. It is well known that flavor and nutritional value are representative indicators of fish quality [2]. In the case of freshwater fish products, consumers tend to prefer products with less flavor and high nutritional value. Although there are several compounds that contribute to favorable flavor, volatile compounds are major contributors to flavor in aquatic products and are thus key factors for consumer acceptance [3,4]. The fish nutritional value indexes mainly include crude protein, crude lipid, fatty acids and amino acids, which are major nutritional components as human food [2]. Therefore, improving flavor quality and nutritional value can effectively enhance the market value of grass carp. The number of attempts to improve the flavor quality and nutritional value of fish is in ascent. It has been reported that dietary plant protein and oil sources significantly reduce the contents of the volatiles and improve the nutritional value in fillets of gilthead sea bream [5]. Low level of dietary fish meal (25%) changes the volatile profiles in cooked fillets of farmed large yellow croaker [6]. The dietary lipid sources (fish oil and krill oil) help to produce more pleasant flavors by increasing the relative levels of 3-methylbutanal, heptanal, benzaldehyde and nonanal in crab muscle, and the diets with marine oil elevate polyunsaturated fatty acids in muscle [7]. In addition to the dietary intervention, water manipulation also affects the flavor of fish. Placing fish in clean water (off-flavor-free) has effectively removed off-odors in various fish species [8]. Several water quality factors (nitrite and ammonia) have been associated with the growth and nutritional value of grass carp [9,10]. However, the effects of water quality factors on flavor quality have seldom been reported. Therefore, further exploration of key water quality factors affecting the flavor quality together with nutritional value will be of great significance for the improvement of freshwater fish quality.

Grass carp culture in China has about 1100 years of history with accumulated farming experience. Now, besides the traditionally-developed commercial feed treatment mode, several novel aquaculture modes have been formed and applied in China, including the faba bean treatment (FBT) mode, grass powder treatment (GPT) mode and waving water treatment aquaculture (WWT) mode. The FBT mode refers to feeding grass carp with faba beans, which leads to higher flesh collagen content and better textual quality [11]. The GPT mode refers to feeding grass carp with grass powder, and the carp cultured by this mode has a bright body surface and sweet flesh [12]. In the WWT mode, grass carp is cultured with traditional commercial feeds and the use of water pumps to make the water surge and gain the advantage of compact flesh [13]. According to practical experience, in the late aquaculture period, both water quality and flesh quality become distinct in these aquaculture modes. Therefore, previous studies exploring the effect of diets on flesh nutritional value in each aquaculture mode should be one-sided. Water quality could be another important factor that accounts for the difference in the flesh flavor quality and nutritional value between these aquaculture modes.

In the present study, combined with the production practice, four typical aquaculture modes in China were simulated: FBT, GPT, WWT and commercial feed treatment (control). We monitored time-dependent changes in water quality parameters during the culture period of 90 days and evaluated flavor quality by volatile flavor compounds detection and an electronic nose analyzer. Nutritional value indexes including crude protein, crude fat, free fatty acids and free amino acids were also determined. Pearson correlation analysis was used to determine the key factors affecting flavor quality and nutritional value.

## 2. Materials and Methods

### 2.1. Fish Culture and Four Treatment Modes

The feeding trial was performed for 90 days in the culture base of Pearl River Fisheries Research Institute. In this study, four typical aquaculture modes were simulated, including the commercial feed treatment (control), faba bean treatment (FBT), grass powder treatment (GPT) and waving water treatment with commercial feed (WWT). The nutritional composition of these experimental diets are shown in Table 1. The fish were cultured in 12 cement pools (2.5 m × 2.5 m × 1.8 m) (three pools for each mode, and 10 fish in each pool). There was no significant difference for the initial weight (838 ± 40 g) between all groups (*p* > 0.05). The fish were fed at 9:00 and 16:00 every day. The daily feeding amount of both commercial feed and faba bean was 200 g and that of grass powder was 400 g. Food residues were collected on the 15 d, 30 d, 45 d, 60 d, 75 d and 90 d, respectively. The total amount of food residues of different diets was as follows: the control with 168 g of residues, FBT with 1620 g of residues, GPT with 890 g of residues and WWT with 183 g of residues. Culture conditions were the same in all pools: water temperature 25–30 °C, pH 6.5–7.5, and the dissolved oxygen > 5.0 mg/L. Generally, in the practical culture process, the frequency of water exchange in grass carp culture ponds is once every 3–4 months. Thus, water in these pools were not exchanged during the culture period (90 days) in the present study.

### 2.2. Detection of Water Quality Parameters

According to a previous study [14], water quality parameters were measured using a microtiter plate reader. Nitrate nitrogen (NO_3_-N) was measured with the phenol disulfonic acid method. Nitrite nitrogen (NO_2_-N) was measured with the Griess–Saltzman method. Total ammonia nitrogen (TAN) concentrations were measured with the spectrophotometric method with salicylic acid. Soluble reactive phosphorus (PO_4_-P) was measured with the molybdenum blue method. Total nitrogen (TN) and total phosphorus (TP) were measured using the potassium persulfate digestion method. All water quality parameters were determined every 15 days during the experimental period.

### 2.3. Sample Collection

After 90 days, the fish were fasted for 24 h, and three fish were sampled from each group. The fish were individually euthanized in pH-buffered tricaine methanesulfonate (250 mg/L) and the body weight and body length were measured. This fish study has a waiver of approval the from ethics committee in conformity with Chinese law. The muscle samples were used to measure volatile aroma components, the contents of water, crude protein and crude fat, free fatty acid and free amino acid profiles. The skin samples were used to determine volatile component. Growth-related parameters were calculated as follows.
Weight gain rate (WGR, %) = (final weight − initial weight)/initial weight × 100
Condition factor (CF, %) = body weight/length^3^ × 100

### 2.4. Determination of Nutritional Value Indexes

In the present study, the methods of the State Standard of the People’s Republic of China (GB5009.4–2016) were used to determine the contents of water (directly drying method), crude protein (Kjedahl method), crude fat (Soxhlet extraction method), free fatty acid contents (acid hydrolysis method) and free amino acid contents (acetyl chloride-methanol and transesterification).

### 2.5. Determination of Volatile Aroma Components

The volatile aroma components of muscle and skin of grass carp were sampled by using solid phase microextraction (SPME) [15]. Volatile compounds were analyzed on a GC–MS apparatus (Shimadzu–QP2010, Kyoto, Japan) operating in the electron ionization mode (EI, 70 eV). The extracted analytes were thermally desorbed by inserting the SPME fiber to the gas chromatograph injector, kept at 250 °C and then separated on a HP-INNOWAX capillary column (30 m × 0.25 mm × 0.25 μm, J&W Scientific Inc., Folsom, CA, USA). The elution program temperatures used during analyses were as follows: (i) 60 °C for 5 min, (ii) increase in temperature with a heating rate of 3.5 °C/min up to 100 °C for 5 min, (iii) increase in temperature with a heating rate of 8 °C/min up to 200 °C for 5 min, and (iv) increase in temperature with a heating rate of 15 °C/min up to 280 °C for 15 min. The carrier gas (He) flow rate was 1.2 mL/min.

Volatile compounds of muscle and skin were identified in the full scan mode (*m/z* 30–550) by NIST mass spectral library (NIST14, version 2.2, National Institute of Standards and Technology, Gaithersburg, MD, USA). Volatile compounds were tentatively identified using the GC–MS spectra. Compounds with ≤ 80% similarity to the NIST library were not considered. In addition, identification was performed by matching their Kovats indices (KI) determined relative to the retention time of a series of n-alkanes (C8–C20) with linear interpolation, with those of authentic compounds or literature data. The chromatographic responses of the detected volatile compounds (peak area counts) were monitored and compared among the studied samples. The relative content of the components were determined with the peak area normalization method.

### 2.6. Analysis of Relative Odor Activity Values

The odor activity value (OAV) reflects the contribution of a single compound to olfaction. This study adopted the relative OAV (ROAV) method, in which the ROAV of less than 1 indicates that the substance has a small contribution to the odor, and a value greater than 1 indicates a significant contribution to the odor [16]. The component making the greatest contribution to the odor of the sample is defined as ROAV*_stan_* and given a value of 100, with the ROAV of other volatile components being calculated as follows:$$ROAV_{i}\approx 100\times \frac{C_{i}}{C_{Stan}}\times \frac{T_{stan}}{T_{i}}$$
where *C_i_* and *T_i_* represent the relative content (the percentage of each compound peak intensity to total peak intensity of all compounds) and the sensory threshold of each volatile component, respectively. *C_stan_* and *T_stan_* represent the relative contents and the sensory thresholds, respectively, of the components contributing most to the total odor of the sample.

### 2.7. Analysis of Flavor Characteristics

According to a previous study [17], 5 g of back muscle and 5 g of skin were collected and homogenized with 2 mL 0.18 g/mL NaCl solution. The mixture was weighed accurately in a 5-mL electronic nose automatic sampling bottle. It was then incubated at 60 °C for 10 min with the cap of the bottle tightened. The clean and dry air was used as the carrier gas at a flow rate of 150 L/min. The injection volume was 2500 μL, injection needle temperature was 55 °C, and data acquisition time was 120 s. The cleaning time of the sensor was 1080 s and 8 fish samples were evaluated in parallel.

### 2.8. Statistical Analysis

All statistical analysis was performed using IBM SPSS Statistics 23 (SPSS Inc., Chicago, IL, USA). Data were analyzed by one-way analysis of variance (ANOVA) followed by Tukey’s test. A *p* value of less than 0.05 was considered as a significant difference. The results were presented as mean ± SE.

Pearson’s correlation was used for correlation analysis between affecting factors (water quality factors, feed) and flavor quality (volatile compounds). Meanwhile, the correlations between affecting factors and nutrients were also analyzed. The feed factors include the amount of eaten feed (feed intake by fish) and uneaten feed (feed residues in water). The clustering algorithm adopted Ward’s minimum variance method. A two-sided *p* value  <  0.05 was considered statistically significant. Correlation analyses were performed by using the open-source Hiplot platform [18].

## 3. Results

### 3.1. Water Quality Analysis

The changes in water quality parameters are shown in Figure 1. All parameters showed an upward trend during the 90 days of the experimental period. The upward trends of total ammonia nitrogen, total nitrogen, soluble reactive phosphorus and total phosphorus were consistent in the four culture modes, while the trends of nitrate and nitrite were different depending on the modes (Figure 1a,b). The accumulations of nitrate and nitrite in the FBT mode were the most significant among the culture modes. At 90 d, the nitrate content in the FBT mode was 9.52 ± 0.23 mg/L, while those of control group, GPT and WWT modes were 4.41 ± 0.07 mg/L, 3.99 ± 0.05 mg/L and 7.18 ± 0.11 mg/L, respectively. Similarly, the nitrite content in the FBT mode increased to 1.08 mg/L at 90 d, while those in other three modes were 0.45 ± 0.039 mg/L (control), 0.15 ± 0.002 mg/L (GPT) and 0.23 ± 0.004 mg/L (WWT), respectively.

### 3.2. Growth Performance and Nutrition Composition

The growth performance of grass carp was significantly different under four modes (Table 2). The control showed the highest weight gain rate and condition factor, while FBT had a lower weight gain rate and condition factor than other treatments. According to Figure 2, the contents of crude protein, crude fat and free fatty acids in the muscle of grass carp were lower in the FBT mode than those in other modes. The water content in the FBT mode was the highest among all modes. The contents of free fatty acids detected were slightly different; the GPT and FBT modes had the highest and lowest contents, respectively (Figure 2d). The different trends for crude fat and free fatty acids between groups are probably because crude fat includes not only free fatty acids but also triglyceride, cholesterol, etc., which were not determined in this study. All amino acids detected showed the lowest content in the FBT mode. Concentrations of some taste-active amino acids, glutamate, aspartate and alanine are shown in Figure 2e, and all detected amino acids are shown in Appendix A.

### 3.3. Analysis of Volatile Compounds

GC-MS was utilized to analyze the volatile compounds in grass carp cultured in the four modes (Appendix A). In the control, the numbers of volatile compounds identified in muscle and skin samples were 29 and 22, respectively. Muscle/skin samples from the FBT, GPT and WWT modes identified 33/35, 24/34 and 30/30 volatile compounds, respectively.

The common volatile compounds include 1-octene-3-ol detected in all tested samples and nonanal and 1-octanol detected in all samples except for the muscle sample of the FBT mode. In the muscle samples of the control (Appendix A), the relative percentage contents of nonanal (23.67%) and 2,4,6-Trimethylmandelic acid (21.12%) were the highest, while in the skin samples, 1-Octen-3-ol, dihydrocoumarin, 4,4,5,7,8-pentamethyl, 1-octanol and nonanal accounted for the highest percentages (8.12–21.11%). In the FBT mode (Appendix A), the relative contents of nonanal, hexanal and hexadecanal were the highest (7.63–17.05%), and those of nonanal, 2-cyclohexen-1-ol and (E)-2-nonenal were the highest in the skin sample (5.96–15.96%). In the muscle samples of the GPT mode (Appendix A), the relative contents of hexadecanal, nonanal and p-isopropoxyaniline were the highest (8.07–20.01%), and those of 1-octen-3-ol, carbamodithioic acid, diethyl-, methyl ester and formamide, N,N-dibutyl- were the highest in the skin sample (7.80–16.70%). Lastly, in the muscle samples of the WWT mode (Appendix A), the relative contents of 2,3-octanedione, hexyl chloroformate, nonanal were the highest (8.44–13.18%), while in the skin sample, the relative contents of the 1-octen-3-ol, carbamodithioic acid, diethyl-, methyl ester and formamide, N,N-dibutyl- were the highest (11.91–24.79%). 

### 3.4. Analysis of Key Flavor Components

The flavor components were further analyzed using the ROAV method, in which a larger ROAV indicates a greater contribution of the components to the overall flavor of the sample. As shown in Table 3, the main flavor components (ROAV > 1) of the control, GPT and WWT modes were similar, while those of the FBT mode were significantly different. In the FBT mode, the main flavor components of muscle sample were nonanal, (E)-2-decenal, 1-octen-3-ol, 1-butanol and hexanal, and the main flavor components of skin sample were (E)-2-nonenal, (E,E)-2,4-decadienal, (E,E)-2,4-nonadienal, (E)-2-decenal, nonanal and octanal. The odor attribute of (E)-2-octenal is fatty, and the odor attribute of (E)-2-nonenal, (E,E)-2,4-decadienal and (E,E)-2,4-nonadienal is a fishy off-flavor, indicating that the main flavor of the flesh in the FBT mode was fishy and fatty. On the other hand, the odors of other modes were mushroom, fatty and grass. These results suggested that grass carp in the FBT mode has the worst odor.

### 3.5. Flavor Characteristics Analysis

The E-nose analysis was used to analyze the flavor characteristics of the flesh samples. The Figure 3 shows the results of principal component analysis (PCA) on the samples from the four modes. The PCA of the four fish muscle samples showed that the sum of the contribution rates of the first (71.84%) and second (26.70%) principal components was 98.54%, which basically covered most of the original information of the samples. The difference between the four modes was mainly attributed to the first principal component in muscle (Figure 3a). Consistent with the results in Table 3, the flavor characteristics of control, GPT and WWT modes were similar. The PCA results of skin sample were consist with those of muscle samples (Figure 3b).

### 3.6. Correlation Analysis

In the present study, we analyzed the correlation between the affecting factors (water parameters and feed (eaten feed and uneaten feed)) and flesh quality. Figure 4a shows that both nitrite (*p* < 0.05, correlation coefficients = 0.95) and nitrate (*p* < 0.05, correlation coefficients = 0.84) had a significant positive correlation with (E,E)-2,4-decadienal, (E,E)-2,4-nonadienal, (E)-2-octenal and (E)-2-decenal, and these four compounds presented fishy odor and fatty odor. On the other hand, uneaten feed (feed residues in water) had a significant positive correlation with nitrite (correlation coefficients = 0.75) and nitrate (correlation coefficients = 1). These results demonstrated that nitrite, nitrate and uneaten feed were tightly connected with flavor quality.

According to Figure 4b, eaten feed (feed intake by fish) had significant positive correlation with crude protein (*p* < 0.05, correlation coefficients = 0.95) and crude fat (*p* < 0.05, correlation coefficients = 0.79). Moreover, eaten feed was positively correlated with the contents of some free amino acids (glutamate, aspartate and alanine). Glutamate and aspartate are known to have an umami taste [23], and alanine has a sweet taste [24]. These results suggested that eaten feed play essential roles in the determination of flesh nutritional value as well as taste characteristics.

## 4. Discussion

Flavor is an important index to evaluate the freshwater fish quality. The flavor quality can be judged according to the odor attribute and threshold value of volatile compounds. The present study demonstrated that the flavor quality of the control, GPT and WWT modes were similar to each other. In these three groups, 1-octen-3-ol, nonanal, hexanal and decanal were the main substances contributing to flavor. It has been reported that 1-octen-3-ol has a mushroom-like odor [25,26]. Nonanal was described to have a fatty aroma [2], and hexanal contributes to the green leafy odor of aquatic products [27]. These volatile substances are common in aquatic products and their odors are generally considered to be pleasant [28,29,30,31]. On the contrary, the odor composition of grass carp in the FBT mode was clearly different from those of other three modes (Table 3; Figure 4). The main flavor contribution substances of the FBT mode were (E)-2-decenal, (E)-2-nonenal, (E,E)-2,4-nonadienal and (E,E)-2,4-decadienal. It has been reported that the odor descriptions of (E)-2-decenal, (E)-2-nonenal are fatty and moss [32], and the odors of (E,E)-2,4-nonadienal and (E,E)-2,4-decadienal have been described as a fishy smell [33]. It is thus likely that grass carp cultured in the FBT mode has a stronger fishy smell compared to those cultured in other three modes.

Taken together with the results of the water quality parameter analysis (Figure 1), it would be possible that grass carp cultured under poor water quality (high nitrate and high nitrite) have a poor flavor quality. Therefore, the regulation of water quality, especially nitrate and nitrite, could be crucial for improving the quality of grass carp. In the future, the actual effects of nitrate and nitrite should be verified by setting different concentration gradients of nitrate and nitrite in the grass carp culture. Moreover, the effect of feed residues (uneaten feed) on water quality cannot be ignored. It has been reported that the accumulation of feed residues in the culture pond led to the increase of content of ammonia nitrogen and nitrite [34]. In addition, the water quality factors such as nitrate and nitrite are related to disease [35,36], immunity [37,38] and metabolism [39,40] and should be carefully monitored in aquaculture.

Diet is the direct cause of the changes in water quality during fish farming. The poor water quality in the FBT mode is likely to be attributed to uneaten faba beans in the water. Furthermore, faba beans may contain proteins that cannot be efficiently metabolized by grass carp, which causes the accumulation of nitrate and nitrite in water. Previous studies have shown that the nitrogen emission rate of fish feeding on faba bean is significantly higher than those of feeding on commercial diets [41]. Interestingly, in the FBT group, the concentrations of nitrate nitrogen (NO_3_-N), nitrite nitrogen (NO_2_-N) and total ammonia nitrogen (TAN) were the highest, but the total nitrogen concentration was the lowest. In this present study, the method for determining total nitrogen is potassium persulfate digestion method [14], and the total nitrogen determined by this method includes total inorganic nitrogen (NO_3_-N, NO_2_-N and TAN) and organic nitrogen. Thus, the reason why the content of total nitrogen in the FBT mode is not the highest could be that the content of organic nitrogen is lower than those in other modes. On the other hand, lower content of nitrogen in the water of the GPT mode was probably because of the low protein content in grass powder. A previous study also found that the water quality is better maintained after feeding on grass powder compared with commercial feed [42]. The overall features of water and flesh quality parameters of the WWT mode were similar to those of control, indicating the importance of diets in these parameters.

It is worth noting that the nutritional composition of grass carp cultured in poor water quality was also the worst in this study. The contents of crude fat and free fatty acids in the FBT mode were lower than other modes, and this may also be related to the low fat content of faba beans [41]. Moreover, the content of glutamate, aspartate and alanine in the FBT mode was lower than those of other modes (Figure 2). These amino acids have been reported to contribute to umami taste [26]. The correlation between the water quality factors and flavor quality (substance for undesirable flavors and amino acids for pleasant taste) substantiated the important roles of nitrate and nitrite in the flavor quality of grass carp.

In conclusion, the present study preliminarily screened the water quality factors that affect the quality of grass carp, highlighting the importance of water nitrite and nitrate as well as eaten feed and uneaten feed. In practical production, adopting technical means to regulate the concentration of nitrate and nitrite in aquaculture water will be a potential method to improve the flavor quality of grass carp. Furthermore, this study demonstrated that feeding administration for eaten feed and uneaten feed would be an effective strategy for the quality improvement of grass carp. These results would provide reference materials for the improvement of flesh quality and consumer acceptance in other freshwater fish.

## Figures and Tables

**Figure 1 foods-10-02075-f001:**
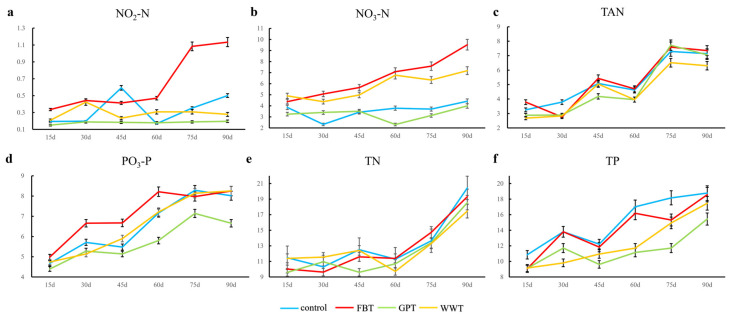
The time-dependent changes of six water quality parameters in the four culture modes. (**a**) NO_2_-N, nitrite nitrogen; (**b**) NO_3_-N, nitrate nitrogen; (**c**) TAN, total ammonia nitrogen; (**d**) PO_4_-P, soluble reactive phosphorus; (**e**) TN, total nitrogen; (**f**) TP, total phosphorus. FBT, faba bean treatment; GPT, grass powder treatment; WWT, waving water treatment with commercial feed.

**Figure 2 foods-10-02075-f002:**
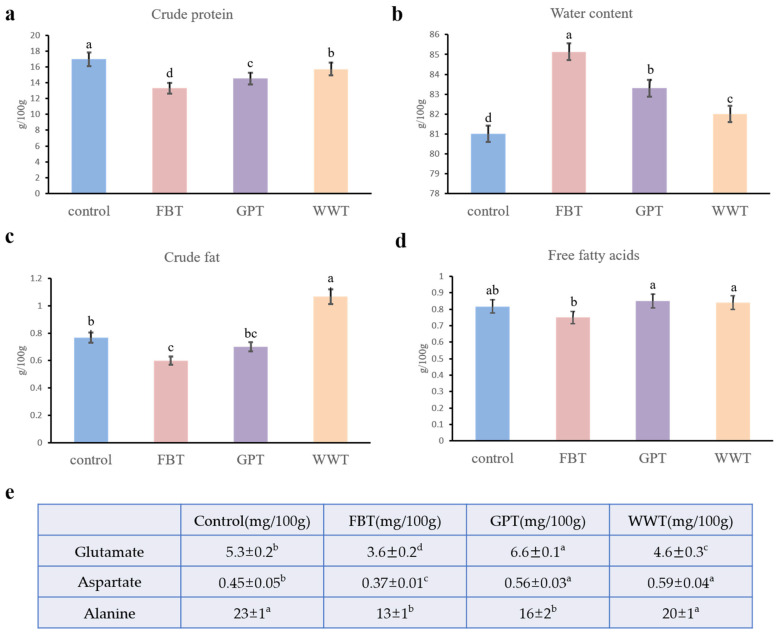
Nutrition composition of grass carp muscle from four culture modes. Statistical analyses were performed using Tukey’s test. FBT, faba bean treatment; GPT, grass powder treatment; WWT, waving water treatment with commercial feed. (**a**) Crude protein; (**b**) Water content; (**c**) Crude fat; (**d**) Free fatty acids; (**e**) free amino acids. Different superscript letters were significantly different (*p* < 0.05).

**Figure 3 foods-10-02075-f003:**
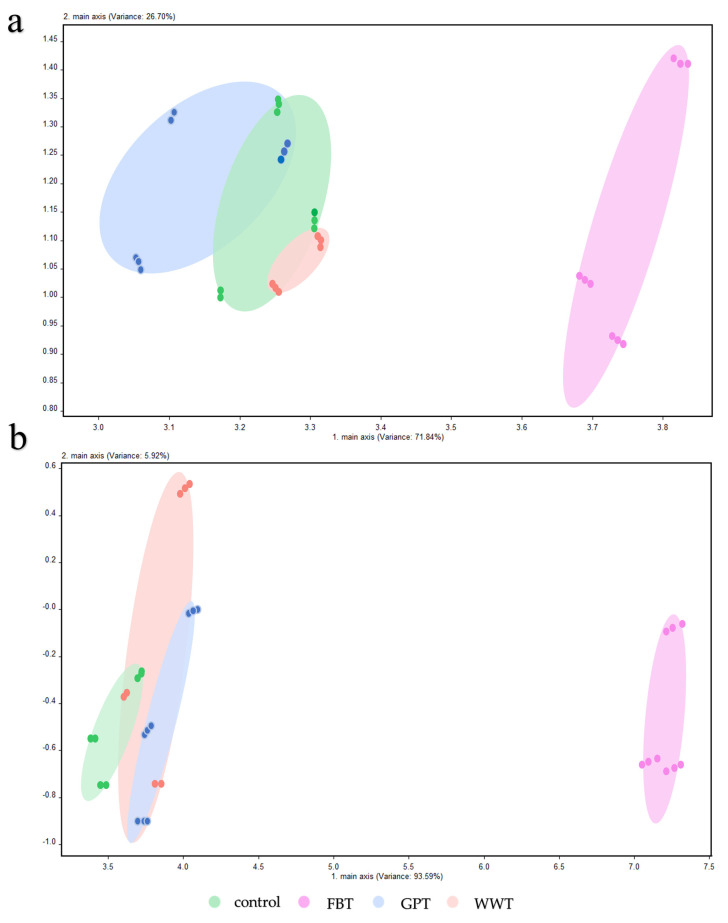
PCA analyses of flavor characteristics from the muscle (**a**) and skin (**b**) samples in the four modes. FBT, faba bean treatment; GPT, grass powder treatment; WWT, waving water treatment with commercial feed.

**Figure 4 foods-10-02075-f004:**
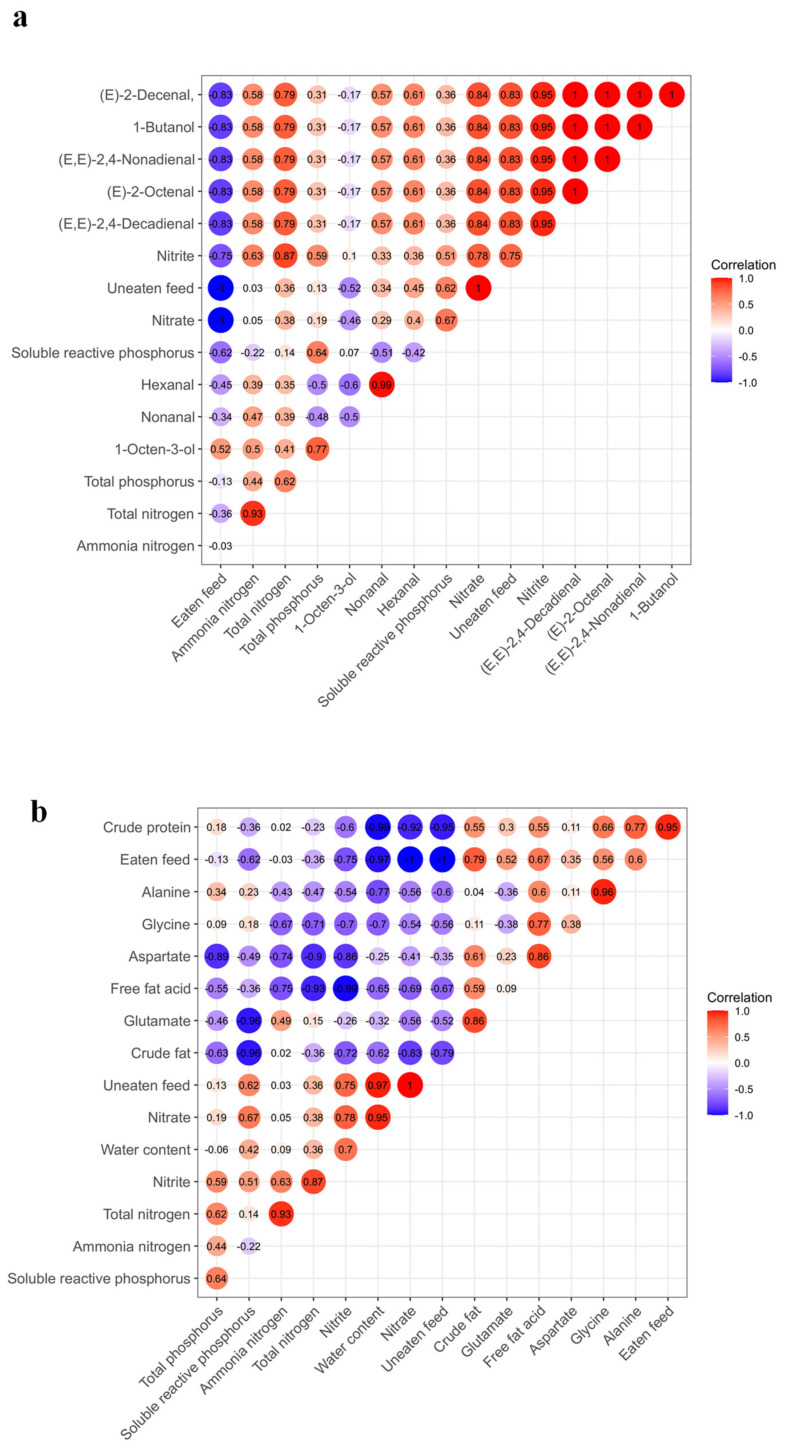
The correlation between flesh quality parameters (main flavor compounds (**a**) and nutritional values (**b**)) and affecting factors (water quality parameters and feed). Red represents a positive correlation, and blue represents a negative correlation. The numbers in the circle represent the correlation coefficient.

**Table 1 foods-10-02075-t001:** Nutritional composition of different diets.

	Crude Protein (g/100 g)	Crude Fat (g/100 g)	Ash (g/100 g)	Moisture (g/100 g)
Commercial diet	28.8	5.5	8.1	10.9
Faba bean	28	1.4	4.1	14.4
Grass powder	14.1	1.8	10.8	12.9

**Table 2 foods-10-02075-t002:** Growth performance of grass carp in different modes.

	Control	FBT	GBT	WWT
Weight gain rate (%)	34.61 ± 0.64 ^a^	7.52 ± 0.73 ^d^	19.62 ± 1.83 ^c^	30.58 ± 1.96 ^b^
Condition factor (%)	33.41 ± 2.7 ^a^	27.18 ± 1.41 ^b^	29.55 ± 0.81 ^a,b^	32.61 ± 3.01 ^a^

Note: Values of the same row with different superscript letters were significantly different.

**Table 3 foods-10-02075-t003:** Main flavor compounds of muscle and skin samples in four culture modes.

Compound	Threshold (μg/kg)	Odor Attributes	Control				FBT				GPT				WWT			
			Relative Content (%)		ROAV		Relative Content (%)		ROAV		Relative Content (%)		ROAV		Relative Content (%)		ROAV	
			Muscle Sample	Skin Sample	Muscle Sample	Skin Sample	Muscle Sample	Skin Sample	Muscle Sample	Skin Sample	Muscle Sample	Skin Sample	Muscle Sample	Skin Sample	Muscle Sample	Skin Sample	Muscle Sample	Skin Sample
1-Octen-3-ol	1.0	Mushroom	0.99	21.11	3.703	100 *	2.40	–	30.572 *	–	2.63	16.17	30.306 *	96.820 *	9.28	24.79	100 *	100
Nonanal	1.0	Fatty	23.67	8.12	88.339 *	38.483 *	5.58	5.18	32.741 *	6.957	8.67	16.70	100 *	100 *	8.44	10.03	90.987 *	40.471 *
Hexanal	4.5	Grass, fresh	–	1.56	–	1.647	17.05	15.96	100 *	21.418 *	7.79	2.92	19.955	3.885	3.14	–	7.520	–
Decanal	0.1	Grass	2.68	–	100 *	–	–	0.66	–	8.800	–	–	–	–	–	0.58	–	23.448
Eucalyptol	3.0	Chemical	1.84	–	2.293	–	–	-	–	–	–	–	–	–	–	–	–	–
Heptanal	3.0	Green	–	–	–	–	–	1.36	–	0.607	–	0.57	–	1.134	–	0.89	–	1.199
Octanal	0.7	Green	–	–	–	–	–	4.22	–	8.099	–	2.39	–	20.407	–	–	–	–
Tetradecanal	110.0	Fishy	–	–	–	–	–	-	–	–	–	–	–	–	2.61	–	0.255	–
(E)-2-Octenal	3.0	Fatty	–	–	–	–	2.48	3.39	4.844	1.515	–	–	–	–	–	–	–	–
(E)-2-Decenal	0.3	Fatty	–	–	–	–	1.86	5.96	36.414 *	26.666 *	–	–	–	–	–	–	–	–
(E)-2-Nonenal	0.08	Fishy, tallow	–	–	–	–	–	5.96	–	100 *	–	–	–	–	–	–	–	–
(E,E)-2,4-Nonadienal	0.090	Fishy, waxy	–	–	–	–	–	2.05	–	30.634 *	–	–	–	–	–	–	–	–
(E,E)-2,4-Decadienal	0.070	Fishy	–	–	–	–	–	4.74	–	90.897 *	–	–	–	–	–	–	–	–

Note: The threshold and odor attributes were found from previous study [2,19,20,21,22]. “–” represents not detected; “*” represents significant figures. Abbreviations: FBT, faba bean treatment; GPT, grass powder treatment; WWT, waving water treatment with commercial feed.

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
