# Peer review of "Key Factors Affecting the Flesh Flavor Quality and the Nutritional Value of Grass Carp in Four Culture Modes"

_foods, 2021, doi:10.3390/foods10092075_

Round 1

Reviewer 1 Report

  In this review manuscript entitled "Key Water Quality Factors Affecting the Flavor Quality of Grass Carp in Four Culture Modes", the authors simulated four typical modes of aquaculture (the commercial feed treatment (control), faba bean treatment (FBT), grass powder treatment (GPT), waving water treatment with commercial feed (WWT)) to investigate the regulation of water quality on the taste of grass carp (Ctenopharyngodon idella), a highly productive freshwater fish. The present study preliminarily screened the water quality factors that affect the quality of grass carp, highlighting the importance of nitrite and nitrate. This result gives interesting information for the research field. I have a few comments, explained below. I hope that my comments are very useful for the improvement of this research.

Comments

(1) Nutritional composition of the diets: Since this journal mainly deals with food, it would be better to show data on the nutritional composition of the feed to grass carp in the control, FBT, and GPT treatment.

(2) Data on food intake and food residues of grass carp should be shown; I think it is necessary to show these data because the water quality in FBT treatment is deteriorating due to food residues.

(3) This is a comment on the grouping of this experiment. The experimental groups in this study are control, FBT, GPT, and WWT. Among these groups, control, FBT, and GPT treatments are feed differently. On the other hand, control and WWT treatments are fed the same feed, but the water flow is different. Therefore, there are two factors in the experimental group: feed and water flow. As far as I could see from the results, there was no significant change in water quality, nutritional composition or volatiles of grass carp between the control and WWT treatments. Therefore, considering the fact that this journal is "Foods", I think it would be better to exclude the data of WWT treatments to make the paper easier to understand. However, I do not feel strongly that the authors must respond to my comment.

(4) 2.4 Determination of nutritional composition: The authors should describe in more detail what fatty acid and amino acid indicate. Are these free fatty acids and free amino acids?

(5) 2.9 Statistical analysis: Authors used the Duncan’s tests as statistical analysis. But, Duncan's multiple range test has been pointed out to have problems such as not taking multiplicity. Thus, please change to another multiple tests.

(6) Figure 1: The NO2-N and NO3-N concentrations in the FBT treatment was higher than the other treatments. On the other hand, there was no significant difference in TN concentration among the groups. Therefore, I think that the authors should consider what component is causing the increase in TN concentration in groups other than the FBT treatment.

(7) Figure 2d: The unit for the y-axis is %, which is inappropriate. It should be shown in g/100g as in other nutrients.

(8) Figure 2e: Please change g/100g to mg/100g.

(9) Table 1: Please review the significant figures of the values in the table.

(10) Discussion: In this experiment, there was a correlation with nitrite and nitrate, and the authors believe that these components reduce the quality of the grass carp. However, it would be difficult to know if nitrite and nitrate actually have an effect without testing. I think it would be better to write about this in the discussion.

Author Response

August 25, 2021

Journal: Foods

Dear reviewer,

Thank you for the comments and recommendations about the manuscript “Key Water Quality Factors Affecting the Flavor Quality and of Grass Carp in Four Culture Modes” (foods-1347913). According to your comments, we have revised the manuscript. Please contact me if you have any questions on this revision. Point-by-point responses are shown below:

Reviewer 1:

In this review manuscript entitled "Key Water Quality Factors Affecting the Flavor Quality of Grass Carp in Four Culture Modes", the authors simulated four typical modes of aquaculture (the commercial feed treatment (control), faba bean treatment (FBT), grass powder treatment (GPT), waving water treatment with commercial feed (WWT)) to investigate the regulation of water quality on the taste of grass carp, a highly productive freshwater fish. The present study preliminarily screened the water quality factors that affect the quality of grass carp, highlighting the importance of nitrite and nitrate. This result gives interesting information for the research field. I have a few comments, explained below. I hope that my comments are very useful for the improvement of this research.

Reply:

Thank you for your kind and constructive comments on our manuscript. According to your comments, we have revised the manuscript as follows. All changes are indicated in the revised manuscript.

Comments

(1)   Nutritional composition of the diets: Since this journal mainly deals with food, it would be better to show data on the nutritional composition of the feed to grass carp in the control, FBT, and GPT treatment.

Reply:

Thank you for your important suggestion. According to your suggestion, we have added the table of nutrition composition of the diets shown in the manuscript as Table 1, which is cited in the line 99.

(2)   Data on food intake and food residues of grass carp should be shown; I think it is necessary to show these data because the water quality in FBT treatment is deteriorating due to food residues.

Reply:

Thank you very much for your comment. We have added the food intake and food residues of grass carp in the four modes to 2.1 Fish culture and four treatment modes of revised manuscript. We also included these data in the correlation analysis (Fig. 4). Lastly, we mentioned the importance of food residues in the second paragraph of Discussion.

(3)   This is a comment on the grouping of this experiment. The experimental groups in this study are control, FBT, GPT, and WWT. Among these groups, control, FBT, and GPT treatments are feed differently. On the other hand, control and WWT treatments are fed the same feed, but the water flow is different. Therefore, there are two factors in the experimental group: feed and water flow. As far as I could see from the results, there was no significant change in water quality, nutritional composition or volatiles of grass carp between the control and WWT treatments. Therefore, considering the fact that this journal is "Foods", I think it would be better to exclude the data of WWT treatments to make the paper easier to understand. However, I do not feel strongly that the authors must respond to my comment.

Reply:

Thank you very much for your valuable advice. It is true that there were no significant differences in water quality, nutritional composition or volatiles of grass carp between the control and WWT treatments. However, we still hope to keep the data of WWT treatments because our data raises a question regarding the value of this commonly used culture mode. As the reviewer kindly expressed the understanding in this comment, the aim of this study is to investigate the key factors in the four culture modes, including the water factors and feed, on the flesh quality. Considering that the flavor is an important factor of grass carp meat as a “food,” our paper should still meet the scope of the journal even if we keep the WWT data.

(4)   2.4 Determination of nutritional composition: The authors should describe in more detail what fatty acid and amino acid indicate. Are these free fatty acids and free amino acids?

Reply:

Thank you for the suggestion. These are free fatty acid and free amino acid contents. We have corrected the descriptions throughout the revised manuscript. Also, we have described the determination of fatty acid and amino acid in more details in the part 2.4 Determination of nutritional value indexes.

(5)   2.9 Statistical analysis: Authors used the Duncan’s tests as statistical analysis. But, Duncan's multiple range test has been pointed out to have problems such as not taking multiplicity. Thus, please change to another multiple tests.

Reply:

Thank you for your precious suggestion. We have changed the analysis method and used Tukey’s test to analyze the data in figure legends and 2.8 Statistical analysis.

(6)   Figure 1: The NO2-N and NO3-N concentrations in the FBT treatment was higher than the other treatments. On the other hand, there was no significant difference in TN concentration among the groups. Therefore, I think that the authors should consider what component is causing the increase in TN concentration in groups other than the FBT treatment.

Reply:

Thank you for your valuable suggestion. We discussed the results of TN, NO2-N and NO3-N concentrations in the FBT group in the third paragraph of the discussion in revised manuscript. Briefly, in this present study, the method for determining total nitrogen is potassium persulfate digestion method [14], and the total nitrogen determined by this method includes total inorganic nitrogen (NO3-N, NO2-N and TAN) and organic nitrogen. So, we think that the content of organic nitrogen in FBT mode could be lower than those in other modes.

(7)   Figure 2d: The unit for the y-axis is %, which is inappropriate. It should be shown in g/100g as in other nutrients.

Reply:

According to your comment, we have changed “%” to “g/100g” in Figure 2d.

(8)   Figure 2e: Please change g/100g to mg/100g.

Reply:

Thank you for the comment. We have changed “g/100g” to “mg/100g” in Figure 2e.

(9)   Table 1: Please review the significant figures of the values in the table.

Reply:

Thank you for your suggestion. We added “*” to Table 3 (Table 1 in the original manuscript) with explanations in the table note.

(10) Discussion: In this experiment, there was a correlation with nitrite and nitrate, and the authors believe that these components reduce the quality of the grass carp. However, it would be difficult to know if nitrite and nitrate actually have an effect without testing. I think it would be better to write about this in the discussion.

Reply:

Thank you very much for this advice. We added the related descriptions in the second paragraph of the discussion in revised manuscript. Indeed, it is difficult to know if nitrite and nitrate actually have an effect without testing. In the future, actual effects of nitrate and nitrite should be verified by setting different concentration gradients of nitrate and nitrite in the grass carp culture.

Sincerely yours,

Ermeng Yu

Pearl River Fisheries Research Institute

Chinese Academy of Fishery Sciences

Guangzhou 510380, China

Reviewer 2 Report

The article deal with the flavour analysis of grass carp farmed in 4 different systems. 

I think that title is not adeguate because the variable of the trial wasn't water quality but the farming sistem and the addition or not of some feed. You didn't act directly on water quality, but only on feeding administration, and only in WWT group acting on water directly. 

 You have to explain bettere the 4 treatment. How many faba bean or grass powder did you used every day? You used these ingredient as replacement of normal feed or in addition?

WWT affect the oxygenation of water? Didn't you analysed this parameter? 

Could you explain in few words the methods of the State Standard of the P.R.C.

Could you provide data of the growing performance of fish? In the rest of article you affirm that Faba bean were not eated by fish, this compomise their growth? 

Fig 2 -d why the unit of measurment in expressed as %?  How do you explain the difference between crude fat and fatty acid? Why you didn't found the same trend intra groups?

-e,  provide SE of Amino Acid determination

Add the SE to data shown in Figure 1. Didn't you evaluated the differences of water parameters between treatments?

Table 1. Provide ritention time and KI of the determined compunds. I think that you missed a lot of volatile compounds. Yuo idetified an averange of 25-30 compunds but you showed less than 10 compounds for group. What about for instance the presence of geosmine? 

How you explain the data of TN? FBT group presented the higher NH4, HO2 and NO3 but wasn't the top of analisys 

Line 299. You mixed 2 different sensory caracteristic that are not correlated. Volatile compound and taste (umami and sweet) involve different sensory and receptor systems.

The stress responce determination in my opinion could be removed from this study because I can not linked with the rest of determinations. 

Line 354. What is the correlation of starch and nitrogenous compound in water? Maybe Faba bean contains some protein content that cannot be metabolised by fish? 

Author Response

August 25, 2021

Journal: Foods

Dear editors and reviewers,

Thank you very much for your valuable comments and recommendations about the manuscript “Key Water Quality Factors Affecting the Flavor Quality and of Grass Carp in Four Culture Modes” (foods-1347913). According to your comments, we have revised the manuscript. Please contact me if you have any questions on this revision. Point-by-point responses are shown below:

Reviewer 2:

The article deal with the flavor analysis of grass carp farmed in 4 different systems. I think that title is not adequate because the variable of the trial wasn't water quality but the farming system and the addition or not of some feed. You didn't act directly on water quality, but only on feeding administration, and only in WWT group acting on water directly.

Reply:

Thank you very much for your kindness and constructive comments on our manuscript. According to your comments, we have changed the title to “Key Factors Affecting the Flesh Flavor Quality and the Nutritional Value of Grass Carp in Four Culture Modes.” Also, we corrected some English expressions in the manuscript. All changes are indicated in the revised manuscript.

(1) You have to explain better the 4 treatment. How many faba bean or grass powder did you used every day? You used these ingredient as replacement of normal feed or in addition?

Reply:

Thank you for your comment. The daily feeding amount of faba bean or grass powder has been added to 2.1 Fish culture and four treatment modes of revised manuscript. These ingredients were used as replacement of normal feed.

(2) WWT affect the oxygenation of water? Didn't you analyze this parameter?

Reply:

Thank you for your comment. We used the water quality analyzer (YSI, USA) to determine the dissolved oxygen in the water and found no difference between the groups. So we did not analyze this parameter. 2.1 Fish culture and four treatment modes describes the absence of difference as “Culture conditions were the same in all pools: water temperature 25 ~ 30 °C, pH 6.5 ~ 7.5, and the dissolved oxygen > 5.0 mg/L.”

(3) Could you explain in few words the methods of the State Standard of the P. R. C.

Reply:

Thank you for your suggestion. We have explained the methods in more details, as shown in the part 2.4 Determination of nutritional value indexes in revised manuscript.

(4) Could you provide data of the growing performance of fish? In the rest of article you affirm that Faba bean were not eaten by fish, this compromise their growth?

Reply:

Thank you for your valuable suggestion. We have added the data of growing performance in grass carp as Table 2 as cited in 3.2 Growth performance and nutrition composition. Indeed, uneaten faba bean likely caused the slow growth of grass carp in this manuscript.

(5) Fig 2-d why the unit of measurement in expressed as %? How do you explain the difference between crude fat and fatty acid? Why you didn't found the same trend intra groups?

Reply:

Thank you for these comments. We have changed “%” to “g/100g” in Figure 2d. Also, for the different trends for crude fat and free fatty acids intra groups, probably because crude fat includes not only free fatty acids but also triglyceride, cholesterol, etc. which were not determined in this study. We have explained these in the revised manuscript, as shown in the part 3.2 Growth performance and Nutrition composition.

(6) -e, provide SE of Amino Acid determination.

Reply:

Thank you for your suggestion. We have added SE of amino acid in Figure 2e.

(7) Add the SE to data shown in Figure 1. Didn't you evaluated the differences of water parameters between treatments?

Reply:

Thank you for your valuable suggestion. We have revised Figure 1 by adding the SE to the data, and the differences was also evaluated.

(8) Table 1. Provide retention time and KI of the determined compounds. I think that you missed a lot of volatile compounds. You identified an average of 25-30 compounds but you showed less than 10 compounds for group. What about for instance the presence of geosmine?

Reply:

Thank you for your valuable suggestion. We are so sorry for that our machine cannot provide KI but it can provide retention time. Because the retention time was different in different samples, we added retention time in supplementary table S2-5. In our manuscript, the revised Table 3 showed volatile compounds which can be searched for threshold and odor attribute, and all volatile compounds were shown in supplementary Tables S2, S3, S4 and S5. Unfortunately, geosmine was not detected by our detection technology. We used normal gas chromatography-mass spectrometry, which is precise enough to detect trace substances such as geosmine.

(9) How you explain the data of TN? FBT group presented the higher NH4, HO2 and NO3 but wasn't the top of analyses.

Reply:

Thank you for your valuable comments.

We discussed the results of TN, NO2-N and NO3-N concentrations in the FBT group in the third paragraph of the discussion in revised manuscript. Briefly, in this present study, the method for determining total nitrogen is potassium persulfate digestion method [14], and the total nitrogen determined by this method includes total inorganic nitrogen (NO3-N, NO2-N and TAN) and organic nitrogen. So, we think that the content of organic nitrogen in FBT mode could be lower than those in other modes.

(10) Line 299. You mixed 2 different sensory characteristic that are not correlated. Volatile compound and taste (umami and sweet) involve different sensory and receptor systems.

Reply:

Thank you very much for this comment. We have separated the volatile compound and taste amino acids during the correlation analysis (Fig. 4a and 4b).

(11) The stress response determination in my opinion could be removed from this study because I can’t link with the rest of determinations.

Reply:

Thank you for your valuable suggestion. We have removed this part from the manuscript.

(12) Line 354. What is the correlation of starch and nitrogenous compound in water? Maybe Faba bean contains some protein content that cannot be metabolized by fish?

Reply:

Thank you very much for these valuable comments. We are so sorry for the wrong expression “the correlation of starch and nitrogenous compound in water”. We have deleted this sentence in the third paragraph of the discussion. Also, we agree with the reviewer's opinion about the protein and added some expressions “Faba bean probably contains some protein content that cannot be metabolized by grass carp” in the third paragraph of the discussion.

Sincerely yours,

Ermeng Yu

Pearl River Fisheries Research Institute

Chinese Academy of Fishery Sciences

Guangzhou 510380, China

Round 2

Reviewer 1 Report

Since my questions and requests were properly replied or revised, I think your manuscript have improved dramatically.